# A novel kinetic energy harvesting system for lifetime deployments of wildlife trackers

Troels Gregersen[1,2,3], Timm A. Wild[2,4,5], Linnea Worsøe Havmøller[1], Peter Rask Møller[1], Torben Anker Lenau[3], Martin Wikelski[2,5,6], Rasmus Worsøe Havmøller[1,2]*

1 Section for Zoology, Natural History Museum of Denmark, University of Copenhagen, Copenhagen, Denmark, 2 Department of Migration, Max Planck Institute of Animal Behavior, Radolfzell, Germany, 3 Section for Engineering Design and Product Development, Department of Civil and Mechanical Engineering, Technical University of Denmark, Kgs. Lyngby, Denmark, 4 Product Development Group Zurich (pd|z), ETH Zürich, Zürich, Switzerland, 5 Centre for the Advanced Study of Collective Behaviour, University of Konstanz, Konstanz, Germany, 6 Department of Biology, University of Konstanz, Konstanz, Germany

* rhavmoeller@ab.mpg.de

**Data Availability Statement:** https://github.com/TroelsG/Kinefox.

**Funding:** LWH received funding from the European Union's Horizon 2020 research and innovation

## Abstract

Wildlife tracking devices are key in obtaining detailed insights on movement, animal migration, natal dispersal, home-ranges, resource use and group dynamics of free-roaming animals. Despite a wide use of such devices, tracking for entire lifetimes is still a considerable challenge for most animals, mainly due to technological limitations. Deploying battery powered wildlife tags on smaller animals is limited by the mass of the devices. Micro-sized devices with solar panels sometimes solve this challenge, however, nocturnal species or animals living under low light conditions render solar cells all but useless. For larger animals, where battery weight can be higher, battery longevity becomes the main challenge. Several studies have proposed solutions to these limitations, including harvesting thermal and kinetic energy on animals. However, these concepts are limited by size and weight. In this study, we used a small, lightweight kinetic energy harvesting unit as the power source for a custom wildlife tracking device to investigate its suitability for lifetime animal tracking. We integrated a Kinetron MSG32 microgenerator and a state-of-the-art lithium-ion capacitor (LIC) into a custom GPS-enabled tracking device that is capable of remotely transmitting data via the Sigfox 'Internet of Things' network. Prototypes were tested on domestic dog (n = 4), wild-roaming Exmoor pony (n = 1) and wisent (n = 1). One of the domestic dogs generated up to 10.04 joules of energy in a day, while the Exmoor pony and wisent generated on average 0.69 joules and 2.38 joules per day, respectively. Our results show a significant difference in energy generation between animal species and mounting method, but also highlight the potential for this technology to be a meaningful advancement in ecological research requiring lifetime tracking of animals. The design of the Kinefox is provided open source.

## Introduction

Tracking wild animals has revealed fascinating insights into migration routes and movement patterns and has become an essential tool for conservationists in protecting everything from

programme under the Marie Skłodowska-Curie
grant agreement No. 801199 (https://rea.ec.
europa.eu/funding-and-grants/horizon-europe-
marie-sklodowska-curie-actions_en). This project
was supported by the Villum Foundation (https://
veluxfoundations.dk/en) through the VILLUM
EXPERIMENT grant no. 36069 awarded to RWH.
The funders had no role in study design, data
collection and analysis, decision to publish, or
preparation of the manuscript.

**Competing interests:** The authors have declared
that no competing interests exist.

individual animals to sites of species level importance [1, 2]. The first attempt at wildlife tracking for scientific purposes can be traced back to the bird-ringing practices of Hans Christian Cornelius Mortensen in the early 1900's, an innovation which paved the way for the revolutionary introduction of VHF wildlife trackers in the 1960s [3].

Continuous technological advancements in energy storage, sensors, and wireless communications technology have since helped to decrease power consumption drastically [4]. However, devices capable of lasting more than a few years, and thus enabling lifetime tracking are still unavailable for most mammalian species. This is mainly due to challenges with the energy supply of mobile tracking devices on free-roaming animals. Today, most wildlife trackers are powered by non-rechargeable lithium-based batteries. The main challenge with batteries is their energy density which, when tracking for multiple years, will add significant weight that makes it impossible to track many smaller species. Arguably, the general consensus currently suggests that wildlife trackers for mammals should not exceed 3–5% of the animal's body weight [5], or it might harm the animals' fitness and movement capabilities [6–10]. Consequently, permits for tracking animals often come with a requirement that devices do not exceed 3–5% of the weight of the tracked animal [11]. Nonetheless, even if tags meet the mass requirements, the possibility of tracking smaller animals for multiple years is limited by battery longevity. Non-rechargeable lithium-based battery technology is advertised to be able to last for up to several decades (e.g., [12]), but studies suggest that wildlife tracking collar malfunctions in many cases are caused by early battery drainage [13], an issue probably caused by a combination of environmental effects (extreme temperatures) and inaccurate battery life estimation (etc. unpredictable energy consumption of GPS).

The challenges regarding weight and longevity of batteries are most often solved with solar cells in combination with rechargeable batteries (e.g., lithium-polymer-based) or supercapacitors. As tracking technology evolved to have a much lower power demand, solar cells gained traction, especially with bird trackers and ear-tags where low weight is critical [14]. Solar cells have become an easy and affordable way to extend the lifetime of a tracking device, but they have their limitations. Few animals live constantly under direct sun, and many animals are nocturnal or live in habitats with a small amount of sun exposure, such as dense rainforests, caves, or merely in low-sunlight habitats at high or low latitudes. Solar cells also suffer from robustness issues, because they require a thin and transparent coating to ensure optimal energy generation [15–17]. To solve some of the challenges related to batteries and solar cells, scientists have attempted to harvest energy using alternative methods. These include a 286 g thermal energy harvesting solution on sheep [18], a 200 g and 20 cm long kinetic energy harvesting solution on reindeer [19] and a 40 g, 6 cm in diameter vibration energy harvester not including casing and collar material [20]. All results are promising but still limited by mass and size.

Humans, on the other hand, have spent the last centuries perfecting energy harvesting in body-worn devices. Automatic watches, also known as self-winding watches, have existed since the late 18th century, and are still widely used. Automatic watches are equipped with a kinetic energy harvester that transforms the wrist's movement into energy used to wind up the watch [21]. These wrist-worn kinetic energy harvesters are lightweight and small enough to fit into a wildlife tracker. These types of harvesters have not previously been used to harvest kinetic energy on animals to power wildlife trackers. In this study we have developed and explored the potential of using a kinetic energy harvesting unit, a 18g, 32mm diameter micro-generator from a smartwatch (Fig 1A) to create a kinetically powered wildlife tracker (the 'Kinefox') for long-term and eventually lifetime tracking of wild animals.

The energy generated from the harvester was stored in a lithium-ion capacitor LIC (Fig 1B). As many wildlife tracking studies are based on collecting and evaluating data from GPS and accelerometers [22], both sensors were integrated into the Kinefox (Fig 1C). The wireless

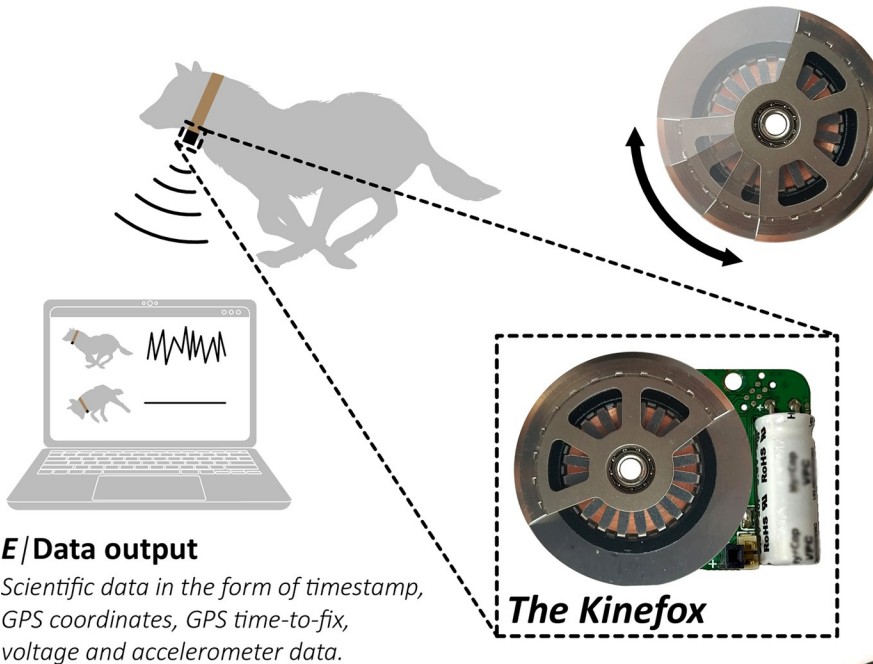

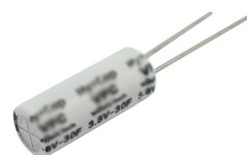

**A | Compact and lightweight energy generation**

*The Kinetron MSG.32 is a small, lightweight energy generator developed for the Sequent Supercharger watch.*

**B | Low leakage, long lifetime energy storage**

*The power generated by the MSG.32 is stored on Lithium Ion Capacitors. LICs have a sigificantly longer lifespan than traditional LiPos.*

**E | Data output**

*Scientific data in the form of timestamp, GPS coordinates, GPS time-to-fix, voltage and accelerometer data.*

**The Kinefox**

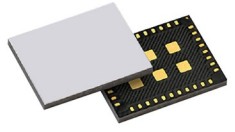

**D | Low-power wide-area networks**

*The collected sensor-data can be wirelessly transmitted over long distances via low-power wireless networks like Sigfox, LoRa and NB-IoT.*

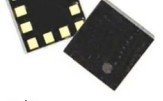

**C | Low-power sensors**

*The stored power can be used to power <uA sensors and state-of-the art ultra low energy GPS/GNSS modules.*

**Fig 1. The Kinefox wildlife tracker concept.** We tested if a lightweight kinetic energy harvester (A), could power an alternative to batteries (B) in the form of a Lithium-Ion Capacitor (LIC). The energy from the LIC was used for low-power sensors (C), including a GPS-module and an accelerometer. Data was then compressed and sent via the low-power wide-area network Sigfox (D). Finally, the data was used to estimate how much energy the harvester generated when mounted on an animals (E).

communication 'Internet of Things' technology Sigfox was integrated for remote data retrieval (Fig 1D). The data were transmitted to an online database for real-time data analysis and energy generation calculations (Fig 1E). Harvesting kinetic energy from animals is only dependent upon the movement of an animal, a behavior that most animals exhibit–at least intermittently—until they die. While lifetime tracking of birds is now a reality due to many decades of development in durable solar tags [23], for many mammal species lifetime tracking is still only a distant reality [24, 25]. The development of a durable kinetically powered light-weight wildlife tracking device would enable lifetime deployments and tracking of mammals. It could, for example, revolutionize studies on mammalian dispersal, a critical life history phase in many mammals when individuals often experience strong natural (mortality) selection for reasons that largely illude scientists. Constant and long-term tracking of individuals will also be a very useful tool in animal-human conflict zones, as well as in conservation and population management in general. The overall goal of this study is to investigate whether the energy generated by the Kinefox prototype, when mounted on animals, is sufficient to daily transmit essential information about an individual via low-power Internet of Things communication technologies.

## Methods

### Ethics statement

All equipment testing on animals were given with the owners' verbal approval and witness by multiple co-authors some of whom where the owners of the dogs used for testing the Kinefox prototypes.

### Electronics

The Kinefox tracker is based on the 18 g, 32 mm diameter micro-generators (MSG32, Kinetron BV, Tilburg, Netherlands) used in the commercially available Sequent Supercharger smartwatch (Fig 2A).

The MSG32 micro-generator is a pendulum-based automatic watch movement, where the pendulum is glued to a ferromagnetic ring. The pendulum with the ferromagnetic ring is placed around a coil of copper wire. When the pendulum swings back and forth, the ferromagnetic ring induces an alternating current in the coil. We integrated a voltage doubler rectifier to transform the alternating current into direct current (Fig 2B). The voltage created by magnetic induction depends on the speed of the magnet and thus the direct current output from the rectifier fluctuates based on the speed of the pendulum. Kinetron BV states that the

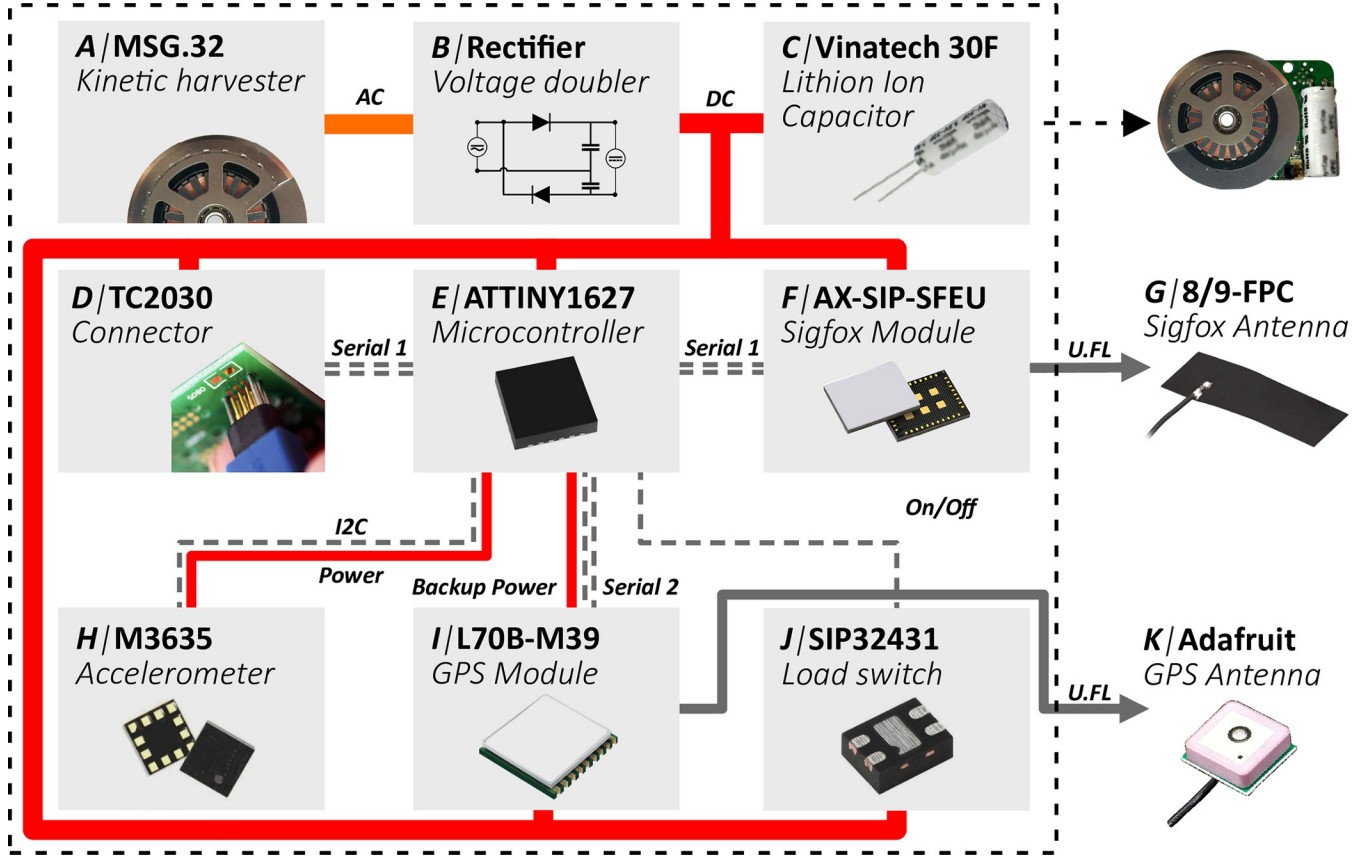

**Fig 2. Hardware overview of the Kinefox.** The MSG32 micro-generator (A) generates an AC current, which is rectified into a DC current by the voltage doubling circuit (B). The current is then used to charge a LIC (C). A Tag-Connector connector (D) is used to charge the LIC as well as program the microcontroller (E). The energy is also used to power a Sigfox module (F), transmitting via a flexible printed circuit board (FPC) antenna (G). The Kinefox V2 also contains an accelerometer (H) and a GPS module (I) which is switched on and off with a load switch (J). The GPS module uses a patch antenna (K).

generator is designed to directly charge a LiPo battery, which assumes a maximum voltage output of 4.2 V, when mounted on a human wrist. For energy storage it was critical to find an alternative to LiPo batteries, due to their poor longevity. A 30 F LIC (VPC, Vinatech) was chosen as this special hybrid supercapacitor is suitable for storing power from rapid high-voltage input, have low self-discharge/leakage and vastly better charge/discharge capabilities [26] than LiPo batteries while still maintaining an acceptable energy density (Fig 2C). Most LiPo batteries are advertised to withstand between 300–500 charge cycles. The selected hybrid supercapacitor advertises 20.000 charge cycles, vastly outperforming LiPo batteries and thus increasing the potential longevity of the Kinefox device. The LIC also has the advantage of a linear relationship between state-of-charge and voltage [27], which makes it possible to estimate energy generation based on voltage change.

The LIC can be charged via a TC2030 Tag-Connect connector (Fig 2D), which is also used to program the microcontroller (ATtiny2627, Microchip technology) (Fig 2E). For remote data transmission we chose Sigfox as the wireless data transmission platform due to its low energy consumption [28, 29], easy integration, and coverage in Denmark and Germany where case studies were conducted. A Sigfox module from ON Semiconductors (AX-SIP-SFEU-1-01-TX30, ON Semiconductors) was selected due to its small footprint (Fig 2F). Furthermore, Sigfox has an integrated location service called Sigfox Atlas Native, which gives a rough estimate of geolocation without requiring additional energy (Triangulation-based). Whip-antennas are traditionally used in wildlife trackers, but they are known to be fragile [30]. Since the goal of the Kinefox is to achieve lifetime operation, internal antennas were explored to maximize longevity. An FPC antenna (8/9-FPC 868, Linx Technologies) (Fig 2G) and a homemade bent-wire monopole antenna made from 1mm cobber wire (S1 Fig) were used for Sigfox. Two hardware versions of the Kinefox were tested in this study. The Kinefox V2 is the one illustrated in the figures throughout this paper. V2 contains the same components as V1, but additionally integrates an accelerometer (MC3635, MEMSIC Inc.) (Fig 2H) and a low-power GPS module (L70B-M39, Quectel) (Quectel) (Fig 2I). Furthermore, the eRIC Sigfox module (eRIC-Sigfox, LPRS) was used in V1 instead of the module from ON-Semi. This is a result of supply-chain issues (performance is almost identical). To keep the power consumption as low as possible, the power to the GPS is turned on and off with a low-power load switch (SIP32431DNP3, Vishay Semiconductors) (Fig 2J). The Adafruit 2461 patch antenna (2461, Adafruit) was used for GPS (Fig 2K). For both Kinefox versions, all the components were placed on a 2-layer printed circuit board (PCB). All design files for both versions of the Kinefox as well as a list of specific components used are open-source available on GitHub (https://github.com/TroelsG/Kinefox).

## Embedded software

Embedded software for the Kinefox was created in C/C++ in Microchip studio and programmed via an Arduino Nano flashed with the jtag2updi firmware [31]. Three firmware versions were used in the experiments. Version 1 and 2.1had the main purpose of sampling and transmitting the voltage of the LIC, used to estimate energy generation, based on the method presented in the Measuring energy generation section. Firmware 2.2 had the main purpose of testing the GPS module and accelerometer, as well as long-term durability and function. All three firmware versions are illustrated as a flowchart in S2 Fig. All firmware versions are based on a loop that can be divided into the following steps: sleep, wakeup, data sampling and transmission. Throughout this paper, these steps will be referred to as a "transmission cycle".

Firmware 1. This firmware was used in the experiments involving all tests with Kinefox V1. In this firmware, the device was coded to wake up and sample the voltage of the LIC. If the

voltage was below 2.9V the device would go to sleep for 4h and continue this cycle until the voltage was above 2.9V. This was to avoid getting below the 2.5V cut-off voltage of the LIC. Four hours was selected as the sleep period, because it was assumed that none of the test animals could generate enough power to overcharge the LIC within this period. If the voltage on the other hand was above 2.9V and below 3.55 V, the firmware code would compress the sampled voltage into 2 bytes in HEX format, transmit via the Sigfox network, go to sleep for 4 hours, and then repeat the transmission cycle. If the voltage was above 3.55 V, the device would go to sleep for only 60 seconds and then repeat the cycle. This was done to quickly lower voltage to avoid overcharging of the LIC. If voltage was above 2.9 V and below 3.55 V every time it woke up, a transmission cycle would be 4 hours long.

Firmware 2.1. This firmware was used for the experiments involving energy generation with Kinefox V2. This firmware is identical to Firmware 1, except that after every transmission cycle, the device goes to sleep 6 times, resulting in a transmission cycle length of 24 hours if voltage was between 2.9 V and 3.55 V at wakeup. After every 4 h sleep period, the device sampled voltage and if was above 3.55 V, the sleep period was lowered to 20 minutes and a transmission cycle would be executed at every wakeup to lower voltage, until the voltage was again above 2.9 V. Furthermore, in this firmware, the 2 byte payload was saved on EEPROM. This way, the data could be downloaded from the device in case of failed transmissions.

Firmware 2.2. The purpose of this firmware version was to test the functionality of the GPS-module and accelerometer. This firmware is identical to Firmware 2.1, only with the addition of what data to transmit. Version 2.1 was coded to sample and transmit 4 seconds of 54 Hz VeDBA sum burst acceleration data (2 bytes), voltage of the LIC (2 bytes), TTF of the GPS fix (1 byte) and a GPS fix (7 bytes), thus transmitting a 12 bytes message.

In addition to the data transmitted by the Kinefox via Sigfox, the timestamp of when the message was received by the Sigfox backend, as well as a sequence number, are part of the metadata of each message. For Firmware 2.1 and 2.2, the Sigfox Atlas positioning data is also included in the metadata.

## Measuring energy generation

The objective of the Kinefox prototypes is to determine how much energy can be produced by the MSG32 micro-generator. To accomplish this, we created a method for measuring the energy generated by the MSG32. This method exploits the approximately linear relationship between the state-of-charge and voltage in LICs. To determine energy generation based on this relationship, we first had to determine a baseline for how much the voltage would drop due to the energy consumption of one transmission cycle without any generated energy.

Two baseline experiments were conducted with the two Kinefox versions with Firmware 1 and 2.1, respectively, which were placed on an office desk until the LIC voltage reached 2.9 V. This resulted in two datasets containing the voltage of the LIC over time, which was fitted with a simple linear regression model with the y-intercept at the origin. The resulting baseline datasets, along with their linear regression lines, are depicted in Figs 4 and 5. The slope coefficient (mV) of these two baseline datasets represents the voltage drop caused by one transmission cycle.

The same experimental setup was employed in the case studies, where we also fitted the data with simple linear regression models. By calculating the difference between the slope coefficient of the linear regression model from the case studies with that of the baseline experiments, we were able to determine the voltage increase caused by the generated energy of the MSG32. Dividing this by the slope coefficient of the baseline experiment allowed us to calculate the number of potential transmission cycles that could be executed with the generated

energy.

$$\text{Generated energy (Number of potential transmision cycles)} = \frac{-\text{Slope coefficient (Baseline)} - \text{Slope coefficient (Casestudy)}}{\text{Slope coefficient (Casestudy)}}$$

The number of potential transmission cycles was converted into joules by multiplying it with the energy consumption of one transmission cycle.

$$\text{Generated energy (Joule)}$$
$$= \frac{-\text{Slope coefficient (Baseline)} - \text{Slopecoefficient (Casestudy)}}{\text{Slope coefficient (Casestudy)}} \cdot \text{Energy consumption for one transmission cycle}$$

## Enclosure

The enclosure for the Kinefox V1 was 3D-printed in PETG plastic (Ultimaker B.V, Utrecht Holland), on a FDM printer (Ultimaker B.V, Utrecht, Holland). In the Kinefox V1, the MSG32 harvester was soldered to the PCB and the enclosure was externally sealed with epoxy. The Kinefox V1 weighed approximately 75 grams.

The enclosure for the Kinefox V2 was printed in CPE+ (Ultimaker B.V, Utrecht Holland). In the Kinefox V2, the PCB (Fig 3A and 3B) had the MSG32 (Fig 3C) mounted with pogo-pins and both parts were then mounted to the enclosure (Fig 3D) with a M3 screw and a heat-set threaded insert.

The enclosure for Kinefox V2 was sealed with a 3D-printed lid and waterproofed with a silicone greased O-ring (Fig 3E). For testing on free-ranging animals, the housing for V2 was additionally sealed with 50 mm diameter heat-shrink and CA-glued tight to the collar (Fig 3F). The total weight of the Kinefox V2 is approximately 150g, varying according to collar size. It

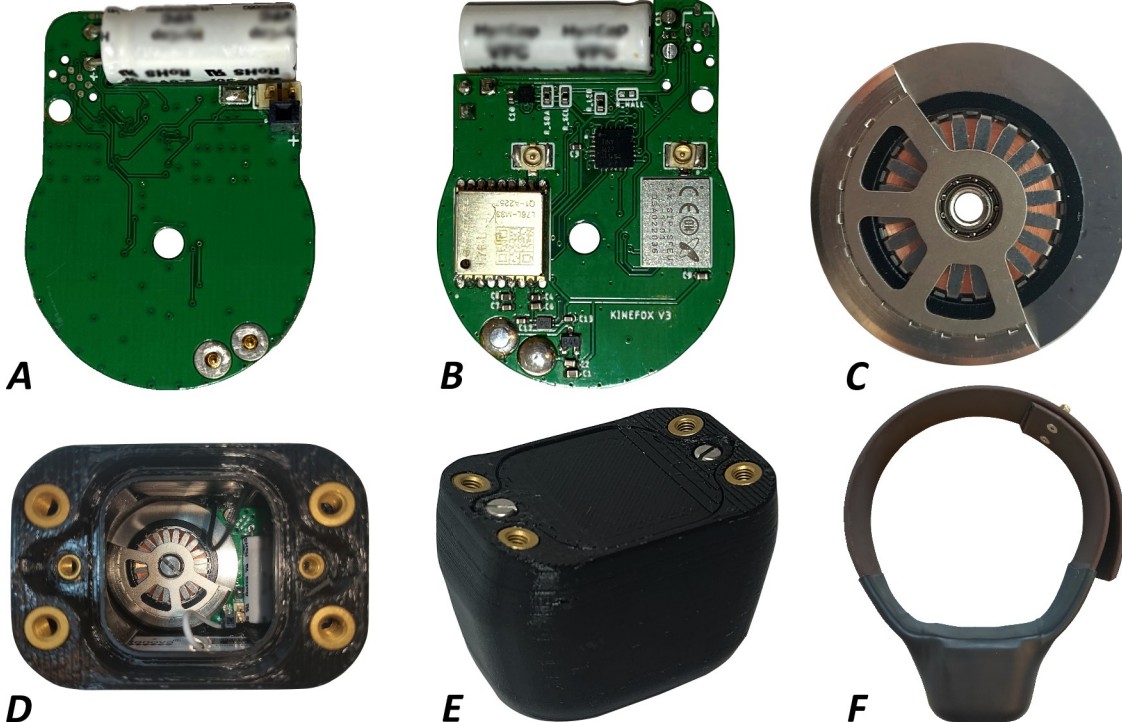

**Fig 3. The Kinefox V2.** A) Top view of the PCB. B) Back view of the PCB. C) The MSG32 micro-generator. D) Top view of the assembled Kinefox placed within the CPE+ printed casing. E) Assembled Kinefox. F) Kinefox mounted in collar made in Biothane. Kinefox and collar is wrapped in heat-shrink for increased waterproofing.

should be noted that weight has not been optimized in this project. Instead, robustness was prioritized. If for example the enclosures were to be injection molded, similar toughness could be achieved with significantly less weight.

## Fittings

In order to investigate the real-world energy generation potential of the Kinetron MSG32, it is important that the collar attachment on the animals resembles "industry standard" as much as possible, as the mounting method can affect the movement of the tracker and thus the energy generation. Collars used for wildlife trackers must last for years and thus be made of a durable material. Collars also need a tight fit against the neck to avoid entanglement with vegetation, rubbing abrasions or falling off. In order to test the Kinefox V1 as early as possible, the devices were mounted in the existing collars and harnesses of the test animals. Based on the learnings from V1, we developed a custom collar (Fig 3F) for the Kinefox V2 based on Biothane (Biothane USA, USA), a nylon-webbing overmolded with PVC plastic. SmartParks use this material for their collar design [32], and we thus knew that it was field-tested to withstand the rigors of wild animals and weather. The Kinefox V2 was bolted to the collar and closed with the same flatheaded bolt and nut system as most conventional wildlife tracking collars [33].

## Case-studies

The Kinefox V1 was tested on two domestic dogs (*Canis lupus familiaris*) (Dog 1 and Dog 2) and a wisent (*Bison bonasus*). For Dog 1, the Kinefox was mounted in its everyday collar (Experiment 1). For Dog 2, two devices were mounted. They were mounted in a harness that the dog was used to, one on the back of the harness (Experiment 2) and one on the chest (Experiment 3). The dog was used to the harness. For the experiments with the dogs, the goal was 6 days of deployment. The Kinefox on the wisent, was mounted to the side of a Vectronics collar (Experiment 4) with the goal of being deployed until device failure. All experiments with the Kinefox V1 was flashed with Firmware 1.

Based on these preliminary results we constructed the complete collar version (Kinefox V2). This prototype was fitted on two domestic dogs (Dog 3 and Dog 4) (Dog 2 and 4 is the same animal) that had their daily walks measured by the owners over a 14-day period to gain insight into how much energy was generated per hour of walking (Experiment 5 and 6). The Kinefox V2 was then mounted on a free-roaming Exmoor pony (*Equus ferus caballus*) for 12 days (Experiment 7). In the experiments with the Exmoor pony, a version with a bend wire antenna for Sigfox communication were used (S1 Fig). In experiment 5–7, the devices were flashed with Firmware 2.1.

No readily available data exist on the TTF of modern GPS modules in wildlife trackers. TTF listed in datasheets of GPS-modules are often bias towards optimal conditions, and do not reflect the conditions of being mounted on a wild animal in their natural habitat. To obtain a rough estimate of TTF on wild animals, the collar fitted to the Exmoor pony was programmed to change to Firmware V2.2 after 12 days (Experiment 8). The goal was to give a rough idea about expected energy consumption by GPS on wild animals, in order to evaluate potential to use the Kinefox for GPS positioning. In addition to GPS-fixes, 4 seconds of VeDBA burst sum acceleration data was sampled at 54 Hz [34], and transmitted within the Sigfox message (12 bytes in total). One of the goals with this experiment was also to test the longevity of the device, so deployment time was set to be until the device stopped functioning. An overview of all the experiments and how they were configured can be seen in Table 1.

**Table 1. Experiments overview.** The table contains all information for each experiment including: firmware version, duration, what animal it was tested on and hardware. In experiments 1–7, the voltage remained between 2.9 V and 3.55 V for the entire duration of the experiment, resulting in regular transmission cycle rates. Experiment 8 generated less energy than it took to execute one transmission cycle per day and thus fell below 2.9 V. After this, the device would only sample and transmit when the voltage surpassed 2.9 V. This resulted in message intervals between 4 days and 24 days, depending on energy generation as well energy consumption by the GPS module as a result of varying TTF.

| Experiment no. | Firmware version | Kinefox Hardware version | Duration | Animal | Data sampled and transmitted | Transmission cycles per day | Fittings | Sigfox antenna | GPS antenna |
|---|---|---|---|---|---|---|---|---|---|
| 1 | 1 | V1 | 6 days | Dog 1 | Voltage of LIC (2 bytes) | 6 | The device was mounted in the everyday collar of the dog. | Linx FPC 8/9 868 | None |
| 2 | 1 | V1 | 6 days | Dog 2 | Voltage of LIC (2 bytes) | 6 | The device was mounted on the chest in the everyday harness of the dog. | Linx FPC 8/9 868 | None |
| 3 | 1 | V1 | 5 days | Dog 2 | Voltage of LIC (2 bytes) | 6 | The device was mounted on the back in the everyday harness of the dog | Linx FPC 8/9 868 | None |
| 4 | 1 | V1 | 17 days | Wisent | Voltage of LIC (2 bytes) | 1 | The device was mounted on the side of a Vectronics collar. | Linx FPC 8/9 868 | None |
| 5 | 2.1 | V2 | 14 days | Dog 3 | Voltage of LIC (2 bytes) | 1 | The device was mounted in a custom collar | Homemade wire antenna | Bentoni (Not used) |
| 6 | 2.1 | V2 | 14 days | Dog 4 | Voltage of LIC (2 bytes) | 1 | The device was mounted in a custom collar | Linx FPC 8/9 868 | Bentoni (Not used) |
| 7 | 2.1 | V2 | 12 days | Exmoor pony | Voltage of LIC (2 bytes) | 1 | The device was mounted in a custom collar | Linx FPC 8/9 868 | Bentoni (Not used) |
| 8 | 2.2 | V2 | 146 days | Exmoor pony | Voltage of LIC, VeDBA sum burst, GPS, TTF (12 bytes) | <1 | The device was mounted in a custom collar. | Homemade wire antenna | Adafruit patch |

## Results

### Power consumption

We measured the average power consumption of the Kinefox V1 and V2 (Table 2) using an Otii Arc (Qoitech AB, Lund, Sweden) power analyzer at 3.3V. The recording of the power consumption can be found in the project GitHub.

The total power consumption per transmission cycle for the Kinefox V1 with Firmware 1 was 0.94 joules. The total power consumption per transmission cycle for the Kinefox V2 with Firmware 2.1 was 1.02 joules. The total power consumption per transmission cycle for the Kinefox V2 with Firmware 2.2 was 4.6 joules (Table 2).

### Case-studies results

The results from experiment 1 showed that Dog 1 with the Kinefox attached to a standard dog collar generated an average of 2.26 joules per day (Fig 4 and Table 3).

The Kinefox mounted on the chest of Dog 2 in experiment 2 generated an average of 10.04 joules per day, while the Kinefox mounted on the back of Dog 2 in experiment 3 generated an average of 3.20 joules per day (Fig 4 and Table 3). The Kinefox was mounted on all three dogs for 6 days, but on the last day of testing, the MSG32 broke in experiment 3 (Ferromagnetic ring broke, could hear it rattling), resulting in only 5 days. The Kinefox on the wisent in

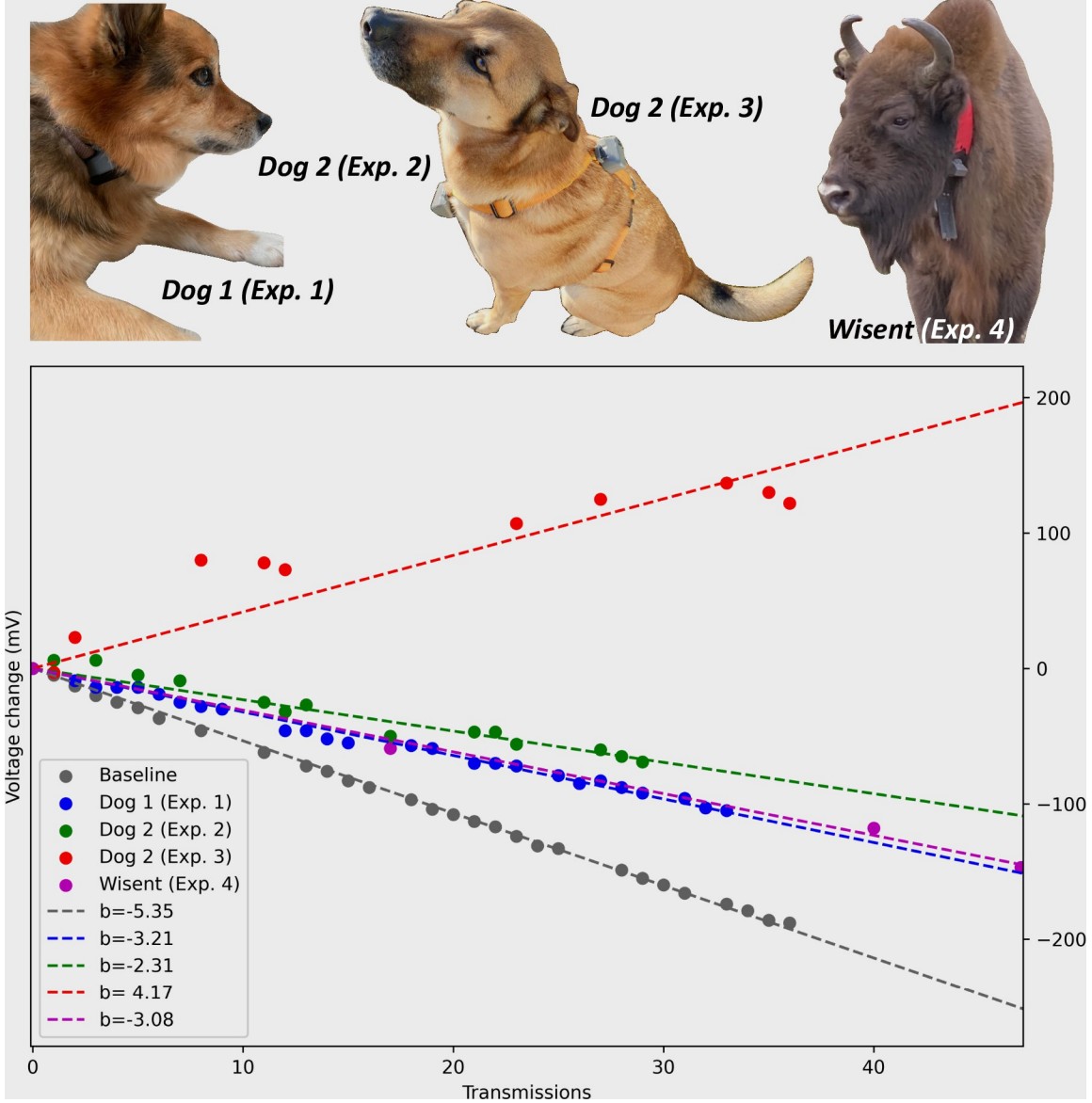

**Fig 4. Chart illustrating energy generation from experiments 1–4.** Experiments were conducted with four different devices on three different animals. Baseline value (grey) shows the decrease in voltage in the LIC over a 9-day period, with no energy generation from the MSG32 and transmitting a 2-byte Sigfox message every 4 hours. Gaps in chart indicate failed Sigfox transmission. Simple regression models have been fitted to all datasets and the slope coefficient is shown in the legend.

experiment 4 generated an average of 2.39 joules per day and survived 17 days (Fig 4), but data is only showed for 8 days. This was due to the fact that the wisent was kept in a small enclosure for the first week and was not representative of normal behavior. Summarized in Table 3, experiments 1–4 generated on average between 2.26 and 10.04 joules per day, which translates into enough energy to execute between 2.4–10.68 transmission cycles. The Kinefox V2 on Dog 3 in experiment 5 generated an average of 1.10 joules per day, when walked an average of 68 min/day (Fig 5 and Table 4).

The Kinefox V2 on Dog 4 in experiment 6 generated and average of 2.25 joules per day, when walked an average of 88 min/day (Fig 5 and Table 4). Finally, the Kinefox V2 on the

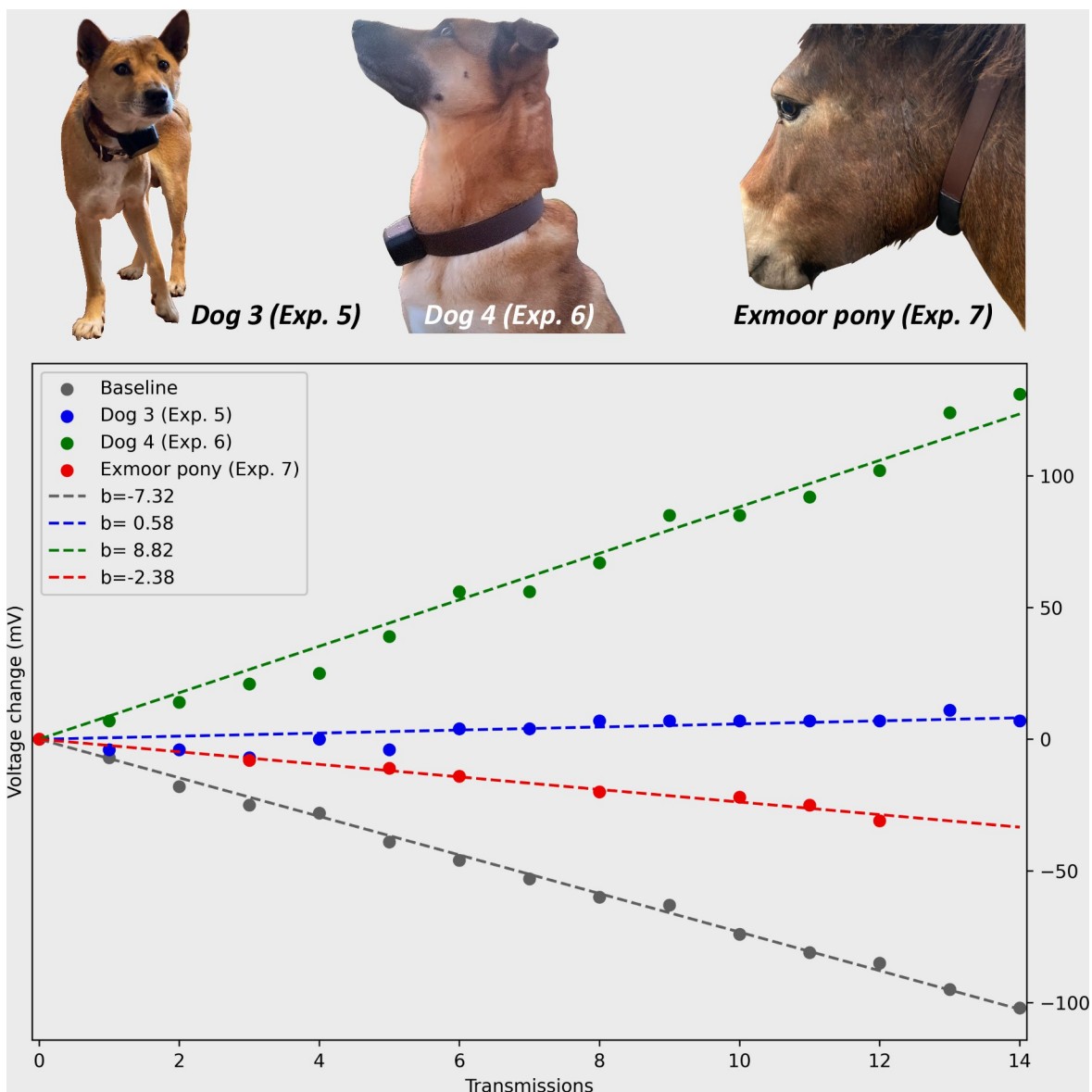

**Fig 5. Chart illustrating energy generation from experiments 5–7.** Experiments were conducted with three different devices on three different animals. Baseline value (grey) shows the decrease in voltage in the LIC over a 14-day period, with no energy generation from the MSG32 and transmitting a 2-byte Sigfox message every 24 hours. Gaps in chart indicate failed Sigfox transmission. Regression models have been fitted to all datasets and the slope coefficient is shown in the legend.

free-roaming Exmoor pony in experiment 7 generated an average of 0.69 joules per day. Overall, the Kinefox V2 generated on average between 0.69–2.25 joules per day, with the Exmoor pony generating the least energy, yet still enough energy to execute on average of 0.67 transmission cycles per day, while Dog 4 generated the most at 2.20 transmission cycles messages per day. The Kinefox was mounted on both dogs for 14 days.

After 12 days on the Exmoor pony, experiment 8 started and the Kinefox V2 started to sample GPS fixes and accelerometer data once a day. At the submission of this paper, a total of 27 GPS fixes were obtained and transmitted, but only 16 received (due to poor antenna performance, not optimized in this study; see Fig 6 for map with GPS points).

**Table 2. Energy consumption overview of Kinefox.** Power consumption recordings were made with the Otii arc power analyzer at 3.3 V. Total energy consumption represent the energy it takes to execute one transmission cycle.

| **Kinefox V1 (Firmware 1)** | | | |
|---|---|---|---|
| Function | Time | Current | Energy consumption (J) |
| Sigfox transmission (2 bytes) | 8.11 s | 31.2 mA | 0.84 |
| Wakeup and LIC voltage sampling | 40 ms | 89.2 uA | $1.18 \cdot 10^{-5}$ |
| Sleep | 4 h | 2 uA | 0.095 |
| Total energy consumption | | | 0.94 |
| **Kinefox V2 (Firmware 2.1)** | | | |
| Sigfox transmission (2 bytes) | 8.17 s | 27.8 mA | 0.75 |
| Wakeup and LIC voltage sampling | 40 ms | 175 uA | $2.31 \cdot 10^{-5}$ |
| Sleep | 24 h | 0.95 uA | 0.27 |
| Total energy consumption | | | 1.02 |
| **Kinefox V2 (Firmware 2.2)** | | | |
| Sigfox transmission (12 bytes) | 10.2 s | 31.3 mA | 1.05 |
| Wakeup and LIC voltage sampling | 40 ms | 175 uA | $2.31 \cdot 10^{-5}$ |
| Sleep | 24 h | 0.95 uA | 0.27 |
| GPS | 59.56s* | 16.6 mA | 3.26 |
| Acc | 4.16 s | 265 uA | 0.0036 |
| Total energy consumption | | | 4.6 |

*GPS Time was the average TTF required for the GPS-fixes in experiment 8. Energy consumption is calculated based on 3.3 V.

The average TTF of these 16 fixes were 59.56 seconds. The power consumption of the GPS module is not constant as it attempts to obtain a GPS fix, but an average was measured to be ~16.6 mA (Table 2). By assuming an average current of 16.6 mA for 59.56 seconds, it required an average of 3.26 joules (at 3.3 V) to obtain a GPS-fix for the Kinefox V2 on the Exmoor pony. The Kinefox V2 on the Exmoor pony reached 2.9V and went to sleep on 2022-12-19. The Kinefox has since woken up and sampled and transmitted 14 GPS fixes. The subsequent intervals between attempts to obtain GPS-fixes and transmit them via Sigfox has been between 4 and 24 days (due to variation in energy generation and GPS TTF).

The accelerometer data confirmed that the Exmoor pony had various levels of activity when it was sampled (S3 Fig) and confirmed that the collar was still fitted on a living animal. The Kinefox on the Exmoor is still ongoing and last transmitted 2023-04-17, 147 days after being deployed.

**Table 3. Experimental results of energy generation from experiments 1–4.**

| | Dog 1 (Exp. 1) | Dog 2 (Exp. 2) | Dog 2 (Exp. 3) | Wisent (Exp. 4) | Baseline |
|---|---|---|---|---|---|
| Slope coefficient (mV)* | -3.21 | 4.17 | -2.31 | -3.08 | -5.35 |
| Generated energy (Number of potential transmission cycles)** | 2.40 | 10.68 | 3.41 | 2.55 | |
| Generated energy (Joule)*** | 2.26 | 10.04 | 3.20 | 2.39 | |

*Slope coefficient is derived from a simple linear regression model fitted to the voltage data transmitted from the Kinefox devices. **Generated energy (Number of potential transmission cycles) is calculated with the method presented in the Measuring energy generation section. Since the transmission cycle for the experiments with the Kinefox V1 was four hours, in order to get the daily generated energy, these values were multiplied by six. ***Generated energy (Joule) was calculated based on the method described in Measuring energy generation section.

**Table 4. Experimental results of energy generation from experiments 5–7.**

|  | Dog 3 (Exp. 5) | Dog 4 (Exp. 6) | Exmoor pony (Exp. 7) | Baseline |
|---|---|---|---|---|
| Slope coefficient (mV)* | 0.58 | 8.82 | -2.38 | -7.32 |
| Generated energy (number of potential transmission cycles)** | 1.08 | 2.20 | 0.67 |  |
| Generated energy (joule)*** | 1.10 | 2.25 | 0.69 |  |

*Slope coefficient was derived from a linear regression model fitted to the voltage data transmitted from the Kinefox devices. **Generated energy (Number of potential transmission cycles) was calculated with the method presented in Measuring energy generation. ***Generated energy (Joule) was calculated based on the method described in Measuring energy generation.

## Discussion

In this study we demonstrate the potential of harvesting kinetic energy of animals to power a wildlife tracker continuously. Using the Kinetron MSG32 micro-generator with state-of-the-art energy storage and low-power electronics, it is demonstrated that it is possible to harvest from 0.69 J on an Exmoor pony to 10.04 J on a domestic dog. The Kinefox V1 on the chest of Dog 2 showed significantly higher values than all in all other experiments. Observations of Dog 2 (Exp. 2) in motion showed that the chest-mounted tracker swung back and forth due to the softness of the harness material. For this reason, a rigid, tighter collar, similar to what is

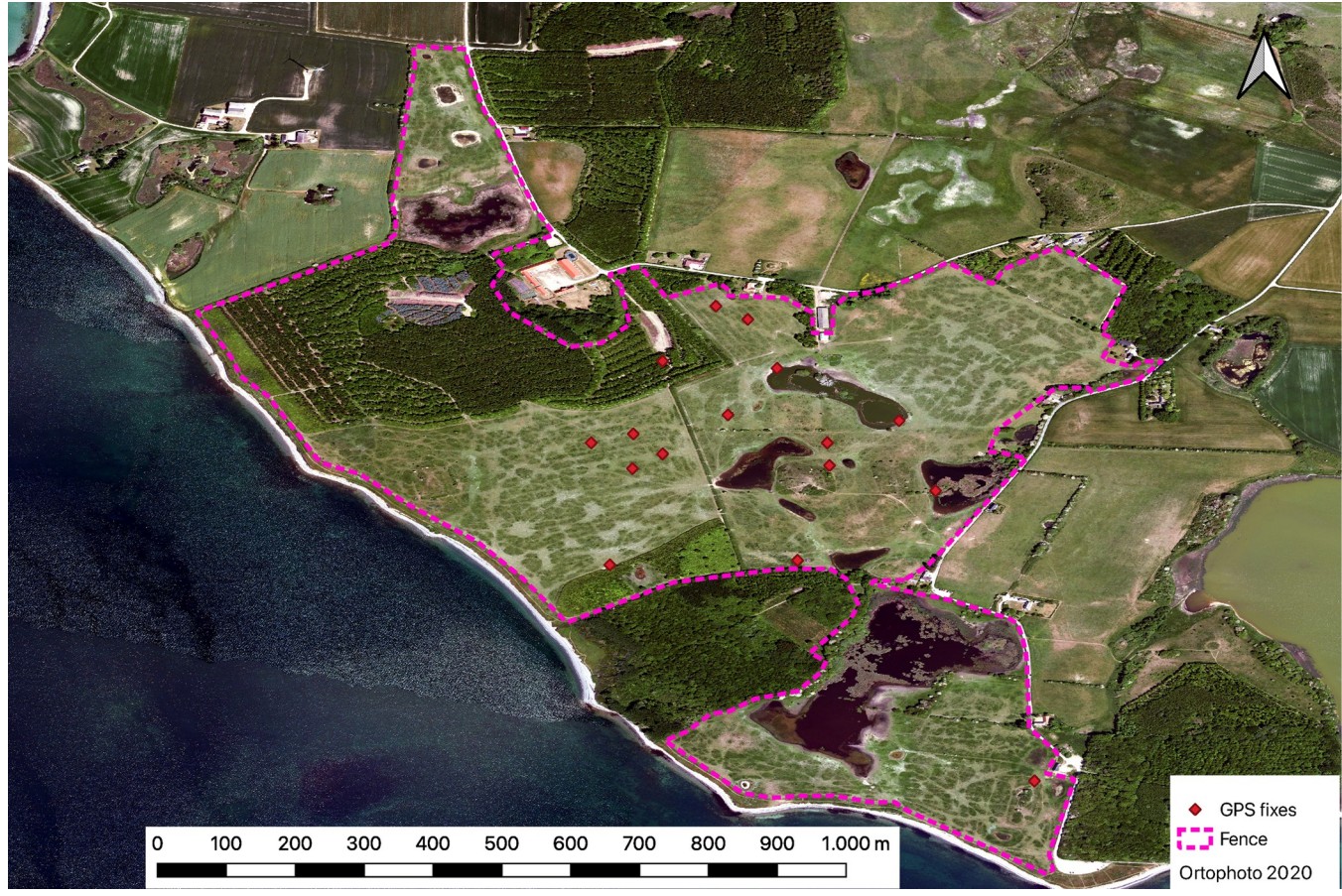

**Fig 6. Map of GPS locations from experiment 8.** Red diamonds indicate GPS-fixes. All GPS-fixes are within the fenced area (dotted magenta line) that the Exmoor pony roams.

used in conventional wildlife tracking collars, was developed for the Kinefox V2. Consequently, the results from Dog 2 (Exp. 2) are not regarded as representative of the potential of the Kinefox.

The aim of this study was to evaluate if kinetic energy could be used to power a wildlife tracker for obtaining lifetime tracks of mammals [24]. In lifetime tracking the least understood phase is often juvenile dispersal, yet it is the most critical for understanding survival and how movement syndromes evolve in mammals [25]. The Kinefox V2 prototype weighs 150 g including collar material, making the Kinefox light enough to be mounted on animals as small as 3 kg when applying the 5% body mass rule [5]. It should be noted that the design of the Kinefox was not primarily optimized for weight at this point, but for robustness instead, and there is great potential for decreasing the weight for use on smaller animals or potentially in an ear-tag. In fact, the weight of the MSG32 harvester itself should be compared to the weight of conventional batteries. In the Kinefox, the MSG32 harvester and a 30 F LIC replaces a conventional battery. These two components weigh a total of 19.9 g. This is roughly equivalent to two non-rechargeable batteries (8.8 g SAFT batteries [35]) that each have a capacity of 1.2Ah (14.255 J at 3.3 V). For a wildlife tracker that uses these two batteries instead of the MSG32 and consumes 2.25 joules per day (2.25 joules is the energy generated on a domestic dog in experiment 6), a quick estimate (2 · 14255 J/ 2.25 J)/365 days) means it will last for 35 years, disregarding self-discharge of the battery. To make the case that the MSG32 is a suitable alternative to these batteries, one or more of the following assumptions must be made.

1. That the MSG32 and the LIC can outlast the 35 years of the batteries.

2. That batteries will discharge significantly due to age or environmental conditions before 35 years have passed.

3. That the MSG32 can generate significantly more energy on wild animals than on domestic animals.

Looking at the first assumption, even if the MSG32 could survive 35 years, it is unplausible that the housing and collar of any wildlife tracker could last this long.

With regards to the second assumption, it is well known that early battery drainage is a huge issue in wildlife tracking [13]. Early battery drainage is most often caused by extreme temperatures, a phenomenon very common in nature. Low temperatures increase resistance in the battery, resulting in lower voltage and thus lower efficiency and higher energy consumption. Overall capacity will also drop due to low temperatures. Oppositely, high temperatures result in increased self-discharge. The battery used in the above example is advertised to have a self-discharge of less than 1% per year [35]. There is however no documentation for exactly how much temperature will affect self-discharge and it is therefore very difficult to verify, since little to no research have been done on these types of batteries (Li-SOCI2). Regardless, without testing, this makes batteries an unreliable energy source for long-term studies.

Considering the third assumption, there is amble evidence that wild animals move more than domestic. The domestic dogs from experiment 6 was walked 1.48 hour per day but were otherwise confined to be indoors. A study by Theuerkauf, Rouys [36] on 11 wolves found that they spend on average 8.6 hours a day moving. If we assume that movement characteristics are similar to the those of the domestic dog in experiment 6, the wolves could generate up to 13.07 joules per day (2.25 J *(8.6 h/1.48 h)). Assuming this, a wildlife tracker powered by batteries would only last for 6 years (under the assumption that the battery tracker powered was also set to consume 13.07 joules per day). This would change the advantage in using MSG32 drastically, assuming the Kinefox and the MSG32 can last for longer than 6 years. For the MSG32 to be superior to the two SAFT batteries, the MSG32 will have to outlive the batteries. This

emphasizes not only the importance in further testing of batteries, but importance of testing the Kinefox prototype in long-term studies on multiple species of wild animals to evaluate the longevity of the MSG32 in real-world scenarios. The MSG32 consists of moving parts and all moving parts experience wear over time. In experiment 3, the ferromagnetic ring broke, a matter that also needs further investigation. It is also worth pointing out that the MSG32 is not specialized for harvesting energy from animals, but from human wrists. Many parameters can be changed to optimize for harvesting on wild animals. The weight of the pendulum, the strength of the magnet and the turns of the copper coil can all be modified.

Maybe, at some point it will be possible to engineer a 5-g harvester that can be used in low-weight ear-tags, as well as on body attachment in birds, fish and small mammals.

Other than optimizing the MSG32 platform itself, there is also potential in exploring placement. In this study, the MSG32 was placed horizontally within the collar (the tracker on the wisent was placed vertically). These were perhaps not the ideal directions to mount the harvester and we suggest that the ideal mounting position varies from species to species and should be optimized. As animal species differ vastly in size and shape as well as movement, it is currently difficult to predict why some species generate more energy than others, but knowing more about which specific movement patterns generate the most energy would make it easier to estimate which animals might be suited for kinetically powered trackers. It would also be interesting to explore the potential in other kinetic harvesting concepts like the spherical harvester developed by Hall and Rashidi [37], which can convert movement to energy from multiple directions.

Nevertheless, even in its current form, the Kinefox might be able to close the gap in wildlife tracking of animals where lifetime data is needed for an ecological study. In order to exemplify the applications of the Kinefox, we present two conceptualized use-cases (Fig 7).

## Long distance dispersal

When conducting ecological studies on animals that disperse hundreds or thousands of kilometers, precision GPS might not be necessary. Examples of long-distance dispersers in Europe are wolves (*Canis lupus*) that have been found to disperse in the excess of 3000 km (Fig 7) [38, 39] and red foxes (*Vulpes vulpes*) that disperse for more than a 1000 km [40]. Knowing that the domestic dogs in this study generated enough energy to transmit 1–2 Sigfox messages a day, it is reasonable to assume that a wild wolf or fox can generate at the least the same amount of energy. And as mentioned previously, it can be estimated that wolves can generate up to 13.07 joules per day, enough to take several GPS-fixes a day (Table 2) [36]. Even if wolves or foxes only generate the same amount of energy as domestic dogs, there is potential for daily positioning. Using the Sigfox Atlas Native location service (requires no additional energy), a rough position with an accuracy between 800 m to 20 km can be achieved. Even though the accuracy of the Sigfox Atlas Native location service is not in the realm of modern GPS locations it would be sufficient to create valuable data about dispersal and migration behavior over not only long distances, but for longer than any existing solutions [29, 41]. Accelerometer data could be used to indicate mortality. Thus, the Kinefox would enable studies on animals that migrate or disperse throughout their entire lives.

## Health monitoring in rewilding projects

Rewilding projects are increasing in numbers around the world, and especially in Denmark these projects are used to bring back nature. One of the challenges in Danish rewilding projects is that they are regulated by the same laws as domestic animals. The legislation concerns the welfare of animals and to prevent die-offs due to starvation as it happened at the Dutch

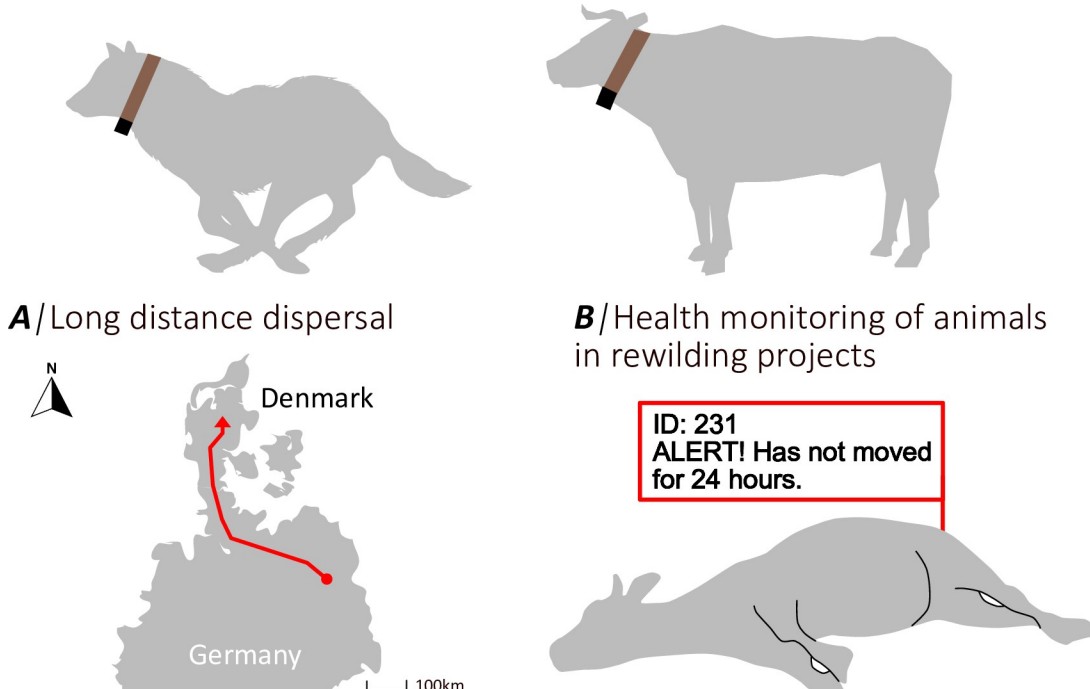

**Fig 7. Potential use-cases of the Kinefox wildlife tracker inspired by real world scenarios.** A) Long distance juvenile dispersal of a wolf from Germany to Denmark. From our results on domestic dogs, it is possible that a wolf moving substantially more would generate enough energy to obtain a daily GPS fix and transmit it via Sigfox. The Sigfox Native Atlas function could serve as an alternative mean of positioning the wolf, though with a larger accuracy error than GPS. B) Monitoring health of animals using accelerometry data. Using VeDBA burst sum, as in experiment 7, it would be possible to study the activity patterns on animals as an indicator for their health and mortality. A sharp change in activity could be an indication that the animal needs attention by caretakes and would serve as a useful monitoring in e.g., a rewilding program.

rewilding project at Oostvaardersplassen. In the winter of 2017/2018 winter, 3.380 deer, horses and cattle died either by natural death or due to starvation because the carrying capacity of the area was surpassed [42]. For managers, this means that the animals must be monitored frequently, a task that can be challenging to do in person. This task could be solved by the Kinefox. Accelerometer data could be sampled several times a day, summarized, and then transmitted once a day via Sigfox (Fig 7). Accelerometry data from rewilding animals fitted with the Kinefox could provide information on how the activity patterns of the animal as an indicator of the health status, as it has been proven useful for in the studies on pigs and cattle by Chapa, Maschat [43] and Riaboff, Shalloo [44]. Four seconds of VeDBA sum burst only requires 0.0036 joules and is sufficient to inform about mortality, thus it would be possible to sample multiple times a day, even with as little energy generation as generated by the Exmoor pony. If for any reason management must attend an animal, an uplink message can be sent to the Kinefox, making it sample and transmit a GPS fix. This might only be necessary to do once a month or less. Even with only 0.69 J generated per day on the Exmoor pony, a GPS fix once a month is possible (as shown in experiment 8). The reason why the Kinefox has an advantage over battery-powered solutions in this case, is that these animals are wild and not handled. Avoiding routinely sedating the animals to replace trackers on the animals is a major advantage.

These examples are based on the results achieved from the tests in this study, but different animals generate different amounts of energy and only the MSG32 micro generator have been tested. To verify the potential of kinetically powered wildlife trackers, continued development

and exploration of the concept should be performed. This is true both in terms of weight, efficiency, and longevity, but also the performance on various species over longer periods of time.

With the Kinefox we have shown that the combination of state-of-the-art electronics and a lightweight kinetic energy harvester can generate enough energy to make a self-powering wildlife-tracker that potentially could last for the entire lifetime of a wild animal.

## Supporting information

**S1 Fig. Picture of the custom bend wire antenna and Adafruit patch antenna inside the Kinefox V2.** This is the version of the Kinefox V2 used in experiment 8 on the Exmoor pony. For the other experiments conducted with the Kinefox V2, the version showed in Fig 3 were used.
(TIF)

**S2 Fig. Flowchart illustration of firmware.** Flowchart illustrating the different firmware versions. Diamond shapes represent conditional statements, while the squares represent functions.
(TIF)

**S3 Fig. Chart with accelerometry data from experiment 8.** Data consists of the average VeDBA sum value sampled over 4 seconds at 54 Hz. The chart shows different levels of activity from low (VeDBA = 349) to high (VeDBA = 16191) probably indicating sleep, as sampling occurred during the dark hours of the night, and high activity in the form of walk/running/feeding respectably. This data confirms that the animal is alive and active.
(TIF)

**S1 File.**
(DOCX)

**S1 Data.**
(ZIP)

## Acknowledgments

Lars Malmborg and Jacob Palsgaard Andersen from Aage V. Jensens Foundation allowed us to attach our prototypes to the GPS collars of their wisents. Naturstyrelsen allowed us fit a collar on one of their Exmoor pony and Marianne Damholt Bergin was incremental in organizing capture and fitting the collars. Shauhin Alavi, Timm A. Wild and Peter Rask Møller allowed us to test prototypes on their dogs; Balto, Pekka and Comet. All three Canids were used to wearing collars.

## Author Contributions

**Conceptualization:** Troels Gregersen, Timm A. Wild, Linnea Worsøe Havmøller, Peter Rask Møller, Martin Wikelski, Rasmus Worsøe Havmøller.

**Data curation:** Troels Gregersen, Timm A. Wild, Peter Rask Møller, Rasmus Worsøe Havmøller.

**Formal analysis:** Troels Gregersen, Timm A. Wild, Rasmus Worsøe Havmøller.

**Funding acquisition:** Rasmus Worsøe Havmøller.

**Investigation:** Troels Gregersen, Timm A. Wild, Martin Wikelski, Rasmus Worsøe Havmøller.

**Methodology:** Troels Gregersen, Linnea Worsøe Havmøller, Torben Anker Lenau, Martin Wikelski, Rasmus Worsøe Havmøller.

**Project administration:** Rasmus Worsøe Havmøller.

**Resources:** Rasmus Worsøe Havmøller.

**Software:** Troels Gregersen, Timm A. Wild.

**Supervision:** Torben Anker Lenau, Rasmus Worsøe Havmøller.

**Validation:** Troels Gregersen, Timm A. Wild, Rasmus Worsøe Havmøller.

**Visualization:** Troels Gregersen, Rasmus Worsøe Havmøller.

**Writing – original draft:** Troels Gregersen, Timm A. Wild, Linnea Worsøe Havmøller, Peter Rask Møller, Torben Anker Lenau, Martin Wikelski, Rasmus Worsøe Havmøller.

**Writing – review & editing:** Troels Gregersen, Timm A. Wild, Linnea Worsøe Havmøller, Peter Rask Møller, Torben Anker Lenau, Martin Wikelski, Rasmus Worsøe Havmøller.

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
