## [Decision Letter · Decision Letter 0]

21 Mar 2023

PONE-D-23-04652A novel kinetic energy harvesting system for long-term deployments of wildlife trackersPLOS ONE

Dear Dr. Havmøller,

Thank you for submitting your manuscript to PLOS ONE. After careful consideration, we feel that it has merit but does not fully meet PLOS ONE’s publication criteria as it currently stands. Therefore, we invite you to submit a revised version of the manuscript that addresses the points raised during the review process.

We look forward to receiving your revised manuscript.

Kind regards,

Ibrahim Sadek, Ph.D.

Academic Editor

PLOS ONE

Journal Requirements:

3. Please ensure that you include a title page within your main document. You should list all authors and all affiliations as per our author instructions and clearly indicate the corresponding author.

Reviewers' comments:

Reviewer's Responses to Questions

**Comments to the Author**

1. Is the manuscript technically sound, and do the data support the conclusions?

Reviewer #1: Yes

Reviewer #2: Yes

Reviewer #3: Yes

2. Has the statistical analysis been performed appropriately and rigorously? 

Reviewer #1: N/A

Reviewer #2: N/A

Reviewer #3: Yes

3. Have the authors made all data underlying the findings in their manuscript fully available?

Reviewer #1: Yes

Reviewer #2: Yes

Reviewer #3: Yes

4. Is the manuscript presented in an intelligible fashion and written in standard English?

Reviewer #1: Yes

Reviewer #2: Yes

Reviewer #3: Yes

5. Review Comments to the Author

Reviewer #1: - This study is interesting where it investigates the use of a small, lightweight kinetic energy harvesting unit as a power source for a custom wildlife tracking device to achieve lifetime animal tracking. The authors integrated a Kinetron MSG32 microgenerator and a lithium-ion capacitor (LIC) into a custom GPS-enabled tracking device capable of transmitting data via the Sigfox 'Internet of Things' network. The prototypes were tested on four domestic dogs, one wild-roaming Exmoor pony, and one wisent. The results showed that enough energy was generated from walking to transmit Sigfox messages with 2 bytes of data per day. The authors conclude that this technology can be a meaningful advancement in ecological research requiring long-term tracking of animals. The design of the Kinefox is provided open source.

- For the supporting figure of the map, I think it would be valuable if included in the main manuscript.

Reviewer #2: In this paper, the authors validate the use of a kinetic energy harvester as a power source for a wildlife tracker. The tracker is equipped with some sensors and a GPS and sends the data via Sigfox. In this work, it is tested on domestic dogs, a pony, and a wisent.

Overall, I find the paper original and I read it with interest. An aspect that should be adjusted is the structure of the paper and the proper allocation of the information in the chapters. A detailed description of the tests, equipment a methodology used, is, for example, missing in the method, a thing that makes the reading complicated.

Introduction, abstract, and discussion should be harmonized. Arguments that explain the unicity of the solution and the reasons why the proposed technology outperforms the existing ones should be addressed more clearly.

Finally, some imprecisions and not clear aspects are present in the results. Below you can find a detailed list of comments.

1. In the introduction, the authors analyze the advantages and disadvantages of alternative energy harvesters to support the choice of the kinetic harvester. They address the robustness issues of solar cells, with which I could agree, however, the robustness of the kinetic harvester is not addressed. Has the Kinetic harvester robustness issues, especially considering that it has mechanically moving parts and that must work outdoors with any temperature and weather?

2. Rows 85-88. Here the authors state that another problem of the solar cell is the reduction of efficiency over time, which makes them not suitable for several years of deployment. This statement is however later contradicted in raw 120-122. I suggest harmonizing the content. Also, I suggest that if performance over years is taken as a drawback for a specific technology, data on this aspect should be offered for the technology suggested in this paper. Is there available information on the aging of the MSG32?

3. Raw 160- 165. How does the max number of charge/discharge cycles of the supercap contribute to the presented device's lifetime?

4. In raw 113-119 the system components are only mentioned, without giving the components' names. Additionally, this part looks out of context in this part. I would suggest moving the text to the chapter “electronics”.

5. In the text, the targeted operation time should be stated clearly and without contradiction. Is the goal a lifetime tracking of the device (raw 36 and 104) or decades (raw 171)?

6. The authors should describe more clearly the differences between prototype V1 and V2 in the chapter electronics. In this chapter, it is clear the additional components in V2 with respect to V1. However, it is not clear which components are integrated into V1. The reader must get to the chapter embedded software before understanding that in v1 only the voltage of the LIC is sent.

7. The authors should integrate a scale in figure 3 so that the reader better understands how compact the device is.

8. The chapter case-study appears to be sound, however, the structure is unclear, making it difficult to follow. I advise the authors to rewrite this part to improve the readability of the text. A table listing the experiments, the prototype used, the period and the animal is helpful to improve the readability

9. The explanation of AVC, ANS, and AGP belongs to the methods, not in the description of table 1. The measurements tests and the parameters measured could be placed in “case-studies”. Additionally, it should be explained how the slope of figure 4 is the voltage change per day if the x-axes of figure 4 are not days but the number of transmissions. The variable AnimalAV is used in table 1 description for the first time without further explanation. To me, it is not clear the definition of this parameter as well as the formula for ANS. About the calculation of AGP, an explanation must be added. Are the authors able to know how many Sigfox messages are generated per day, or only the received messages are counted? Given that some messages get lost, by using the received messages, the energy consumption would be underestimated.

10. Raw 315-317. Could you please give here more details? Does the Kinefox still operate even if in sleep mode?

11. The measurement of energy consumption is shown and explained in the chapter “power consumption”. However, it is already used in the previous chapter (formulas in Tables 1 and 2). I suggest that the chapter “power consumption” is placed before the first use of measured values.

12. Part 328-334. Could the authors define what information was sent in the 2Bytes message, as they did for the 12-bytes message? Useful for the reader would be here to have a schematic representation of the payload.

13. For the power consumption experiments, the tests (instrumentations, protocol, etc.) must be described in the chapter Method.

14. In the discussion, it is addressed the difference between the collar materials used in prototype V1 and V2. This aspect must be addressed more clearly in the methods. As said in point 6 of this list a better explanation of the two prototypes is necessary for better comprehension.

15. Part 365- 383. The authors should explain in a more scientific way how and when the MSG32 outperforms batteries. The authors for example assert the necessity of their solution if 20J are used per day. Can the Authors explain, why exactly this value (which by the way was also not achieved in their work)? The Authors should make an energy comparison considering the features and energy consumption of devices on the market or from the literature.

16. The assertion in 381-383 is not supported by literature

17. The references should be checked. For some articles, the list of authors is not complete.

18. The text contains spelling errors (for example raw 73, 240, 241)

Reviewer #3: The idea is new and there is an innovation in it.

The structure is good.

The study method provides the purpose of the study.

The conclusion section is good.

The English is good/

The Idea and methodology are excellent.

6. PLOS authors have the option to publish the peer review history of their article (what does this mean?). If published, this will include your full peer review and any attached files.

Reviewer #1: **Yes: **Hassan M. Ahmed

Reviewer #2: No

Reviewer #3: No

---

## [Author Response · Author response to Decision Letter 0]

28 Apr 2023

Dear Dr. Ibrahim Sadek

Thank you very much for the opportunity to submit a revision of our manuscript. We have worked through the comments of the reviewers, mainly reviewer 2, and have done our best to address the concerns raised. In short, we have updated and included the of map of the GPS-locations of the wild-roaming Exmoor pony as suggested by reviewer 1. We have addressed all of the issues raised by reviewer 2, where the most significant changes are re-arrangement of the methods section, clarifications and added details on how the firmware code works as well as specifications on how the experiments were set up.

We hope that you find our work sufficient for publication.

Reviewer #1: - This study is interesting where it investigates the use of a small, lightweight kinetic energy harvesting unit as a power source for a custom wildlife tracking device to achieve lifetime animal tracking. The authors integrated a Kinetron MSG32 microgenerator and a lithium-ion capacitor (LIC) into a custom GPS-enabled tracking device capable of transmitting data via the Sigfox 'Internet of Things' network. The prototypes were tested on four domestic dogs, one wild-roaming Exmoor pony, and one wisent. The results showed that enough energy was generated from walking to transmit Sigfox messages with 2 bytes of data per day. The authors conclude that this technology can be a meaningful advancement in ecological research requiring long-term tracking of animals. The design of the Kinefox is provided open source.

- For the supporting figure of the map, I think it would be valuable if included in the main manuscript.

Response to reviewer 1

Response: We thank the reviewers for the positive feedback and have incorporated the map into the main manuscript as suggested

Reviewer 2

In this paper, the authors validate the use of a kinetic energy harvester as a power source for a wildlife tracker. The tracker is equipped with some sensors and a GPS and sends the data via Sigfox. In this work, it is tested on domestic dogs, a pony, and a wisent. 

Overall, I find the paper original and I read it with interest. An aspect that should be adjusted is the structure of the paper and the proper allocation of the information in the chapters. A detailed description of the tests, equipment a methodology used, is, for example, missing in the method, a thing that makes the reading complicated. 

Introduction, abstract, and discussion should be harmonized. Arguments that explain the unicity of the solution and the reasons why the proposed technology outperforms the existing ones should be addressed more clearly. 

Finally, some imprecisions and not clear aspects are present in the results. Below you can find a detailed list of comments. 

1. In the introduction, the authors analyze the advantages and disadvantages of alternative energy harvesters to support the choice of the kinetic harvester. They address the robustness issues of solar cells, with which I could agree, however, the robustness of the kinetic harvester is not addressed. Has the Kinetic harvester robustness issues, especially considering that it has mechanically moving parts and that must work outdoors with any temperature and weather?

Response: The purpose of the introduction is to present the issues with current technology. We address the issues on the robustness etc. of a kinetic harvester in the discussion. 

2. Rows 85-88. Here the authors state that another problem of the solar cell is the reduction of efficiency over time, which makes them not suitable for several years of deployment. This statement is however later contradicted in raw 120-122. I suggest harmonizing the content. Also, I suggest that if performance over years is taken as a drawback for a specific technology, data on this aspect should be offered for the technology suggested in this paper. Is there available information on the aging of the MSG32? 

Response: The statement about the efficiency of the solar cells have been deleted and the content has been harmonized. Unfortunately, there is no data available on the aging wear of the MSG32, which adds to the need of long-term testing of kinetic harvesters.

3. Raw 160- 165. How does the max number of charge/discharge cycles of the supercap contribute to the presented device's lifetime? 

Response: This section about charge/discharge cycles have been expanded to be more specific. 

4. In raw 113-119 the system components are only mentioned, without giving the components' names. Additionally, this part looks out of context in this part. I would suggest moving the text to the chapter “electronics”. 

Response: A more detailed overview of the parts are included in the chapter “electronics”, but we think it makes sense to already mention the components in the introduction, to present to concept. 

5. In the text, the targeted operation time should be stated clearly and without contradiction. Is the goal a lifetime tracking of the device (raw 36 and 104) or decades (raw 171)?

Response: We thank the reviewer for the suggestion and have changed it to lifetime. 

6. The authors should describe more clearly the differences between prototype V1 and V2 in the chapter electronics. In this chapter, it is clear the additional components in V2 with respect to V1. However, it is not clear which components are integrated into V1. The reader must get to the chapter embedded software before understanding that in v1 only the voltage of the LIC is sent.

Response: To make the difference between V1 and V2 clearer, the “Embedded software” section have been rewritten to more clearly describe the difference between the firmware versions. Furthermore, a table (Table 1) have been included that shows how the different hardware and firmware versions are configured in the experiments. 

7. The authors should integrate a scale in figure 3 so that the reader better understands how compact the device is. 

Response: We appreciate the suggestion, but the pictures in figure 3 are not equally sized, so a scale would not work. The relevant size limitation in the device is the harvester itself, which has its dimensions clearly stated in the “electronics” chapter. All other measurements can be derived from the projects GitHub.

8. The chapter case-study appears to be sound, however, the structure is unclear, making it difficult to follow. I advise the authors to rewrite this part to improve the readability of the text. A table listing the experiments, the prototype used, the period and the animal is helpful to improve the readability 

Response: Table 2 have been added to clearly show the experiments and how they were configured. The experiments listed in the table are clearly referenced throughout.

9. The explanation of AVC, ANS, and AGP belongs to the methods, not in the description of table 1. The measurements tests and the parameters measured could be placed in “case-studies”. Additionally, it should be explained how the slope of figure 4 is the voltage change per day if the x-axes of figure 4 are not days but the number of transmissions. The variable AnimalAV is used in table 1 description for the first time without further explanation. To me, it is not clear the definition of this parameter as well as the formula for ANS. About the calculation of AGP, an explanation must be added. Are the authors able to know how many Sigfox messages are generated per day, or only the received messages are counted? Given that some messages get lost, by using the received messages, the energy consumption would be underestimated. 

Response: A method section “Measuring energy generation” have been added that describes this method for calculating energy generation. 

Regarding the point if only the received messages are counted, a sequence number is transmitted along with the Sigfox message. This makes it possible to include calculations for messages that were not received. 

10. Raw 315-317. Could you please give here more details? Does the Kinefox still operate even if in sleep mode? 

Response: The “embedded software” chapter have been expanded to describe how the firmware works in more detail. Furthermore, a supplementary figure (S2) have been made that illustrated how the firmware functions.

11. The measurement of energy consumption is shown and explained in the chapter “power consumption”. However, it is already used in the previous chapter (formulas in Tables 1 and 2). I suggest that the chapter “power consumption” is placed before the first use of measured values.

Response: We thank the reviewer for the suggestion and have moved the power consumption section to the start of the results section. 

12. Part 328-334. Could the authors define what information was sent in the 2Bytes message, as they did for the 12-bytes message? Useful for the reader would be here to have a schematic representation of the payload. 

Response: The “Embedded firmware” chapter has been expanded and now clearly states what the payload contains. Is also shown in the new experiments overview table (Table 2). 

13. For the power consumption experiments, the tests (instrumentations, protocol, etc.) must be described in the chapter Method. 

Response: These power consumption measurements are very simple and straightforward, thus there is no further details to be described than what is already in the methods chapter. 

14. In the discussion, it is addressed the difference between the collar materials used in prototype V1 and V2. This aspect must be addressed more clearly in the methods. As said in point 6 of this list a better explanation of the two prototypes is necessary for better comprehension.

Response: The difference between the versions have been expanded in both the “embedded software” section, the new experimental overview table 2 as well as the slightly expanded “Fittings” section. 

15. Part 365- 383. The authors should explain in a more scientific way how and when the MSG32 outperforms batteries. The authors for example assert the necessity of their solution if 20J are used per day. Can the Authors explain, why exactly this value (which by the way was also not achieved in their work)? The Authors should make an energy comparison considering the features and energy consumption of devices on the market or from the literature.

Response: The section in the discussion about the comparison to batteries have been detailed to more clearly show how the MSG32 compares to batteries and which assumptions still need to be confirmed in order to validate the MSG32 as an alternative to batteries. Little to no literature exists on power consumption on devices already on the market and the data that exists is often on relatively old technology. We have selected the Sigfox wireless technology due to its extremely low energy consumption and Sigfox is only just starting to being used in wildlife tracking. The same is true for the GPS-module we use. For this reason, comparing to market devices would not make sense. 

16. The assertion in 381-383 is not supported by literature

Response: We appreciate the reviewer picking up on this and we have changed the wording for clarification. 

17. The references should be checked. For some articles, the list of authors is not complete. 

Response: the reference list has been double checked and is according to the Vancouver style that PloS One uses. This means for publications with more than seven authors, the first seven are mentioned and the remaining are referred to as “et. al.” as per the submission guidelines.

18. The text contains spelling errors (for example raw 73, 240, 241)

Response: Thank you for noticing. We have fixed the spelling errors. 

Response to reviewer 3

Reviewer #3: The idea is new and there is an innovation in it.

 The structure is good.

 The study method provides the purpose of the study.

 The conclusion section is good.

 The English is good/

 The Idea and methodology are excellent.

---

## [Decision Letter · Decision Letter 1]

5 May 2023

A novel kinetic energy harvesting system for lifetime deployments of wildlife trackers

PONE-D-23-04652R1

Dear Dr. Havmøller,

We’re pleased to inform you that your manuscript has been judged scientifically suitable for publication and will be formally accepted for publication once it meets all outstanding technical requirements.

Kind regards,

Ibrahim Sadek, Ph.D.

Academic Editor

PLOS ONE

Additional Editor Comments (optional):

Reviewers' comments:

Reviewer's Responses to Questions

**Comments to the Author**

1. If the authors have adequately addressed your comments raised in a previous round of review and you feel that this manuscript is now acceptable for publication, you may indicate that here to bypass the “Comments to the Author” section, enter your conflict of interest statement in the “Confidential to Editor” section, and submit your "Accept" recommendation.

Reviewer #2: All comments have been addressed

2. Is the manuscript technically sound, and do the data support the conclusions?

Reviewer #2: Yes

3. Has the statistical analysis been performed appropriately and rigorously? 

Reviewer #2: N/A

4. Have the authors made all data underlying the findings in their manuscript fully available?

Reviewer #2: Yes

5. Is the manuscript presented in an intelligible fashion and written in standard English?

Reviewer #2: Yes

6. Review Comments to the Author

Reviewer #2: (No Response)

7. PLOS authors have the option to publish the peer review history of their article (what does this mean?). If published, this will include your full peer review and any attached files.

Reviewer #2: No

---

## [Editor Report · Acceptance letter]

9 May 2023

PONE-D-23-04652R1 

A novel kinetic energy harvesting system for lifetime deployments of wildlife trackers 

Dear Dr. Havmøller:

I'm pleased to inform you that your manuscript has been deemed suitable for publication in PLOS ONE. Congratulations! Your manuscript is now with our production department. 

Kind regards, 

on behalf of

Dr. Ibrahim Sadek 

Academic Editor

PLOS ONE